# Verification of *AKT* and *CDK5* Gene and RNA Interference Combined with Irradiation to Mediate Fertility Changes in *Plutella xylostella* (Linnaeus)

**DOI:** 10.3390/ijms25094623

**Published:** 2024-04-24

**Authors:** Jiaqi Wen, Mengran Wang, Yuhao Zeng, Fengting He, Shifan Li, Ke Zhang, Qunfang Weng

**Affiliations:** 1College of Plant Protection, South China Agricultural University, Guangzhou 510642, China; wjiaqi168@163.com (J.W.); andolfwang@163.com (M.W.); 13533061123@163.com (Y.Z.); 18824094422@163.com (F.H.); lishifan0308@163.com (S.L.); 2Key Laboratory of Bio-Pesticide Innovation and Application of Guangdong Province, College of Plant Protection, South China Agricultural University, Guangzhou 510642, China

**Keywords:** irradiation, RNA interference, *Plutella xylostella*, *AKT* gene, *CDK5* gene, fertility

## Abstract

*Plutella xylostella* (Linnaeus) mainly damages cruciferous crops and causes huge economic losses. Presently, chemical pesticides dominate its control, but prolonged use has led to the development of high resistance. In contrast, the sterile insect technique provides a preventive and control method to avoid the development of resistance. We discovered two genes related to the reproduction of *Plutella xylostella* and investigated the efficacy of combining irradiation with RNA interference for pest management. The results demonstrate that after injecting *PxAKT* and *PxCDK5*, there was a significant decrease of 28.06% and 25.64% in egg production, and a decrease of 19.09% and 15.35% in the hatching rate compared to the control. The ratio of eupyrene sperm bundles to apyrene sperm bundles also decreased. *PxAKT* and *PxCDK5* were identified as pivotal genes influencing male reproductive processes. We established a dose-response relationship for irradiation (0–200 Gy and 200–400 Gy) and derived the irradiation dose equivalent to RNA interference targeting *PxAKT* and *PxCDK5*. Combining RNA interference with low-dose irradiation achieved a sub-sterile effect on *Plutella xylostella*, surpassing either irradiation or RNA interference alone. This study enhances our understanding of the genes associated with the reproduction of *Plutella xylostella* and proposes a novel approach for pest management by combining irradiation and RNA interference.

## 1. Introduction

The diamondback moth (*Plutella xylostella* (Linnaeus)) is one of the most globally destructive and economically impactful pests of cruciferous vegetables [1], causing great losses in cruciferous plant production [2,3,4]. *Plutella xylostella* poses a significant threat to the foliage of vegetables during its larval period, with adults laying eggs on the fresh leaves, leading to significant economic losses due to diminished yields and compromised produce quality [4,5,6]. In addition, due to the short life cycle of *Plutella xylostella* and the intensive use of chemical pesticides in the field, *Plutella xylostella* has developed resistance to most pesticides [7,8,9,10,11,12]. Therefore, an environmentally friendly and sustainable method and strategy for *Plutella xylostella* control is urgently needed.

The sterile insect technique (SIT), based on irradiation, is a method to control pest populations. It involves the mass production of insects, the sterilization of males through irradiation [13,14,15] and the release of these sterilized males into wild areas, where sterile males mate with wild females, resulting in no offspring and thereby reducing pest populations [16,17,18]. This technique has been applied to Lepidoptera pests and has shown good control effects [19], such as with *Thaumatotibia leucotreta* (Meyrick) [20], *Teia anartoides* [21], *Cactoblastis cactorum* (Berg) [22] and *Cydia pomonella* [23]. In the aforementioned examples, the lepidopteran pests were effectively suppressed or eradicated. In this technique, two main doses are commonly utilized: a complete-sterility dose and a sub-sterility dose. Insects exposed to the complete-sterility dose through irradiation are rendered incapable of producing offspring, and insects exposed to the sub-sterile dose can produce offspring, but the offspring will be sterile [14,15,24]. Insects treated with the sub-sterility dose exhibit a significant enhancement in mating competitiveness compared to those exposed to the complete-sterility dose, thereby increasing the population control effect [15,25]. Consequently, using the sub-sterile dose to induce insect sterility offers higher efficiency and longer-lasting effects in practical operations, while also being easier to implement. According to our previous research, the sub-sterile dose of ^60^Co-γ irradiation for *Plutella xylostella* was determined to be 200 Gy [26].

RNA interference is now also used as a new method of preventing and controlling pests through gene editing techniques, usually by feeding or injecting dsRNA into insects to alter the expression levels of essential genes to induce toxic effects [27]. RNA interference has also shown good control effects against some Lepidopteran pests [28]. The growth of larvae of *Helicoverpa armigera* was slowed when they were fed plants that expressed double-stranded RNA (dsRNA) specific to *CYP6AE14* [29]. After silencing the *Pxylinx4* of *Plutella xylostella* females, it was found that the number of eggs laid was significantly reduced [30]. Additionally, after silencing *Tssor-3* and *Tssor-4*, both the number of eggs laid and the hatching rate were significantly reduced [31].

Currently, irradiation sterile insect technology is often complemented by additional techniques, resulting in enhanced efficacy in pest control compared to the use of a single technology [32]. Therefore, the integration of RNA interference with irradiation represents a promising avenue worth exploring, potentially offering a more efficient and effective approach to induce the sterility of pests and to regulate pest populations.

Our group discovered that ^60^Co-γ radiation can induce specific genetic effects in *Plutella xylostella,* resulting in a sub-sterile effect in males at 200 Gy and a complete-sterile effect in males at 400 Gy. To further investigate these effects, proteomics technology and bioinformatics analyses were used to explore alterations in the gene expression levels within the male testes of *Plutella xylostella* following exposure to ^60^Co-γ radiation [33]. In the present study, nine significantly differentially expressed genes in the testes of *Plutella xylostella* after irradiation treatment were verified, including *nuclear factor NF-kappa-B (NFkB)*, *cyclin-dependent-like kinase-5 (CDK5)*, *3-phosphoinositide-dependent protein kinase-1 (PDK1)*, *inhibitor of nuclear factor kappa-B kinase subunit alpha (IKKα)*, *phosphatidylinositol 4, 5-bisphosphate 3-kinase (PI3K)*, *serine/threonine—protein kinase mTOR—like (mTOR)*, *phosphatase and tensin homolog (PTEN)*, *RAC serine/threonine-protein kinase (AKT)* and *tyrosine—protein kinase hopscotch (JAK2)* [34]. We employed double-stranded RNA (dsRNA) injections as a gene knockdown approach to confirm that irradiation reduces the reproductive ability of *Plutella xylostella*. Through this approach, we screened out two genes, *PxAKT* and *PxCDK5*, that significantly affect reproduction in response to irradiation. Moreover, we investigated the effects of RNA interference combined with irradiation on the reproduction and life parameters of *Plutella xylostella*. We posit that this combined technology holds substantial reference and practical significance for controlling the *Plutella xylostella* population by inducing sterility.

## 2. Results

### 2.1. The Effects on the Expression Levels of Nine Genes of Irradiation

In the present study, nine significantly differentially expressed genes (*PxPI3K*, *PxJAK2*, *PxmTOR*, *PxPTEN*, *PxAKT*, *PxNFkB*, *PxCDK5*, *PxPDK1* and *PxIKKα*) were chosen for the test [23]. The six-day-old male pupae of *Plutella xylostella* were irradiated with 0 Gy, 200 Gy and 400 Gy, and the testes of *Plutella xylostella* were dissected and taken out 24 h after emergence. The expression levels of the genes in the testes were characterized using qRT-PCR.

As shown in Figure 1, the results show that nine genes were significantly highly expressed with 400 Gy treatment. Notably, the relative expression levels of *PxAKT* and *PxPI3K* were up-regulated the most, reaching 16.87-fold and 13.85-fold higher levels than those observed in the control group. The relative expression levels of nine genes exhibited significant reductions following irradiation at 200 Gy. Notably, the relative expression levels of *PxCDK5* and *PxAKT* experienced the most substantial decreases, declining by 62.37% and 57.67% compared to the control group. It can be seen that the expression levels of nine genes in the testes were affected by 200 Gy and 400 Gy irradiation.

### 2.2. The Effects on Plutella xylostella Reproductive Indicators of Gene-Knockdown

As shown in Figure 2, after injection with dsRNA in male pupae, the mRNA expression levels of nine genes in males after emergence for 24 h were significantly suppressed compared to both the DEPC water-injected group and the blank control group (CK).

After the parental males were injected with ds*PxAKT* and ds*PxCDK5*, the number of eggs laid by the F1 generation significantly decreased by 28.06% and 25.64% (Figure 3A), and the hatching rate of the eggs also significantly decreased by 19.09% and 15.35% (Figure 3B).

### 2.3. The Effects of PxAKT and PxCDK5 Knockdown on Plutella xylostella Sperm Bundles

Six-day-old male pupae were injected with ds*PxAKT* and ds*PxCDK5* and dissected 24 h after emergence. The results show that injection with ds*PxAKT* and ds*PxCDK5* had no significant effect on the weight of the testes and the total number of sperm bundles in males (Figure 4A). However, the number of eupyrene sperm bundles in the treatment group injected with ds*PxAKT* and ds*PxCDK5* reduced by 18.72% and 15.42% compared with the control group, and the number of apyrene sperm bundles in the treatment group injected with ds*PxAKT* and ds*PxCDK5* increased by 50.95% and 22.39% compared with the control group (Figure 4B). Injection with ds*PxAKT* and ds*PxCDK5* significantly reduced the ratio of eupyrene sperm bundles to apyrene sperm bundles.

### 2.4. The Alterations in PxAKT and PxCDK5 Expression Levels and in the Fertility of Plutella xylostella across Generations Following Irradiation Treatment

*Plutella xylostella* were irradiated with a sub-sterile dose of 200 Gy. The changes in the expression levels of *PxAKT* and *PxCDK5*, as well as alterations in the fertility of *Plutella xylostella* across the F0–F4 generations, were detected.

As shown in Figure 5, it was observed that the expression levels of *PxAKT* and *PxCDK5*, as well as the fertility of *Plutella xylostella*, gradually recovered over several generations of reproduction. Notably, the recovery patterns of the expression levels of the two genes and the fertility were essentially the same. This implies that changes in the expression levels of *PxAKT* and *PxCDK5* induced by irradiation may exert an influence on the reproductive capacity of male *Plutella xylostella*, potentially manifesting as genetic effects.

Six-day-old male pupae were irradiated at 0 Gy, 50 Gy, 100 Gy, 150 Gy, 200 Gy, 250 Gy, 300 Gy, 350 Gy and 400 Gy, and were then dissected 24 h later. We established a dose–effect relationship for the expression levels of *PxAKT* and *PxCDK5*. The results show that there was a consistent decline in the expression levels of *PxAKT* and *PxCDK5* from 0 Gy to 200 Gy. However, beyond 200 Gy, there was a gradual increase in the expression levels of *PxAKT* and *PxCDK5* from 200 Gy to 400 Gy. The fitting function for the relative expression level of *PxAKT* within the range of 0 Gy to 200 Gy is represented as y = −0.0027x + 0.994, r^2^ = 0.98. The fitting function for the relative expression level of *PxAKT* within the range of 200 Gy to 400 Gy is described as y = 0.008e^0.0186x^, r^2^ = 0.96. The fitting function for the relative expression level of *PxCDK5* within the range of 0 Gy to 200 Gy is represented as y = −0.0029x + 0.998, r^2^ = 0.98. The fitting function for the relative expression level of *PxCDK5* within the range of 200 Gy to 400 Gy is described as y = 0.0067x − 0.934, r^2^ = 0.98, where “y” represents the expression level of the gene, “x” represents the dose of irradiation and “r” represents the degree of correlation.

### 2.5. The Impact of PxAKT Injection Combined with Low-Dose Irradiation on Plutella xylostella Fertility

The expression level after ds*PxAKT* injection was equal to the expression level after 135 Gy irradiation treatment, as demonstrated by the results of the calculation based on the formula fitted in Section 2.4 and the expression level following ds*PxAKT* injection in Section 2.2. Using the sub-sterility of *Plutella xylostella* as the standard for pest management, we chose to combine *PxAKT* knockdown with 65 Gy irradiation to achieve sub-sterile irradiation dosage levels.

As shown in Figure 6, after subjecting mature pupae to 65 Gy of ^60^Co irradiation, there was a significant reduction in the adult lifespan and emergence rate of the parental generation, a significant decrease in the pupation rate of the F1 generation and a significant decrease in the number of eggs laid and the hatching rate of the F2 generation. However, the reproductive ability of *Plutella xylostella* essentially returned to pre-irradiation levels by the F4–F5 generation.

As shown in Figure 6, after the mature pupae were injected with ds*PxAKT*, there was a significant reduction in the adult lifespan and the emergence rate of the parental generation, as well as a significant reduction in the pupation rate, the number of eggs laid and the hatchability of the F1 generation. However, measuring the reproductive markers of the following F1–F3 generations revealed that just the injection of ds*PxAKT* had a brief effect on the growth, development and reproduction of *Plutella xylostella*. The adult lifespan and the emergence rate essentially reverted to their pre-treatment values by the F1–F2 generation, and the number of eggs laid and the rate of hatching essentially reverted to their pre-treatment levels by the F2–F3 generation.

As shown in Figure 6, after low-dose irradiation combined with ds*PxAKT*-injected treatment, *Plutella xylostella* experienced a severe reduction in growth, development and reproductive capacity, with slow recovery afterwards. The growth, development and reproductive capacity of *Plutella xylostella* did not return to normal pre-treatment levels until the F5–F6 generation.

The results demonstrate that, of the three treatment approaches, the combination of ds*PxAKT* injection and low-dose irradiation affects *Plutella xylostella* fertility more permanently and requires more generations for fertility to recover.

### 2.6. The Effects of Low-Dose Irradiation Combined with PxCDK5 Knockdown on the Fertility of Plutella xylostella

The expression level after ds*PxCDK5* injection equates to the expression level after 89 Gy irradiation treatment, based on the formula fitted in Section 2.4 and the expression level after ds*PxAKT* injection in Section 2.2. Using a standard dose of 200 Gy, which results in sub-sterility in *Plutella xylostella*, we decided to attain a sub-sterile irradiation dose level by combining *PxAKT* injections with 111 Gy irradiation.

As shown in Figure 7, after low-dose irradiation, the adult lifespan of the F0 generation was the lowest, having decreased by 36.15% compared with the control group. The F2 generation had the lowest number of eggs laid and the lowest hatching rate, decreasing by 29.27% and 36.96% compared with the control group. The F1 generation had the lowest emergence rate and the lowest pupation rate, decreasing by 25.92% and 37.19% compared with the control group. It is evident that the growth, development and fertility of *Plutella xylostella* were significantly impacted by low-dose irradiation, and the fertility of these populations could not be fully recovered until the F5 generation.

As shown in Figure 7, after the injection of ds*PxAKT*, the adult lifespan and the emergence rate of the F0 generation were significantly lower than those of the control group, falling by 16.04% and 16.00%. The number of eggs produced, the rate of hatching and the rate of pupation of the F1 generation were significantly lower than those of the control group, falling by 13.58%, 17.73% and 11.84%. The growth, development and fecundity of *Plutella xylostella* were all significantly impacted by the injection of ds*PxCDK5*, but this effect was temporary, and the fecundity of the population was entirely recovered by the F2 generation.

As shown in Figure 7, after low-dose irradiation combined with ds*PxCDK5* injection, the F0 generation emergence rate was 31.17% lower compared to the control group. The F1 generation exhibited the lowest rates of pupation, adult longevity and egg production, with respective reductions of 42.10%, 41.51% and 32.98% compared to the control group. The hatchability of the F2 generation was the lowest, having decreased by 34.18% when compared to the control group. The effect on the fertility of male *Plutella xylostella* was long-lasting, which is attributed to the combination of low-dose irradiation and ds*PxCDK5* injection. The fertility of the population was not entirely recovered until the F6 generation.

These results demonstrate that, of the three treatment approaches, the combination of ds*PxCDK5* injection and low-dose irradiation affects *Plutella xylostella* fertility more permanently and requires more generations for fertility to recover.

## 3. Discussion

This study employed RNA interference (RNAi) to validate the genes associated with male fertility in *Plutella xylostella* and investigated the synergistic effects of combining irradiation with silencing of the reproductive genes. Previous research has primarily concentrated on elucidating the repercussions of irradiation on the physiology and fertility of *Plutella xylostella*. Although transcriptomic and proteomic analyses have hinted at various genes and pathways potentially influencing the reproductive processes of *Plutella xylostella*, the precise mechanisms remain elusive. Furthermore, prior investigations have predominantly focused on the singular impact of irradiation technology on *Plutella xylostella* and have not yet explored its integration with other technological approaches.

The present study used RNA interference to confirm the genes linked to the male fertility of *Plutella xylostella*, and the results suggest that the *PxAKT* and *PxCDK5* genes have a significant impact on male *Plutella xylostella* fertility. This is consistent with the results of previous studies on other species. A previous study found that the *AKT* gene is an important part of the *PI3K-AKT* signaling pathway, which is closely related to the development of germ cells and the reproductive behavior of *Plutella xylostella* [35], *Bombyx mori*, *Aedes aegypti* [36], *Spodoptera litura* and humans [37]. Phosphorylated tyrosine proteins play an important role in reproductive processes, such as participating in sperm production and sperm capacitation, as well as participation in the acrosome reaction during fertilization [38,39,40]. In our research, it was found that when the *PxAKT* gene of male *Plutella xylostella* was knocked down, fertility was significantly affected. This phenomenon has also been verified in studies on other insects. After the *AKT* genes of female of *Chrysopa pallens* [41], *Lasioderma serricorne* [42] and *Frankliniella occidentalis* [43] were knocked down, fertility was significantly reduced, indicating that this gene may indeed be involved in the regulation of some reproductive functions. Previous studies have found that the *CDK5* gene is involved in the regulation of a variety of reproductive-related proteins and plays an important role in the differentiation and apoptosis of germ cells [44,45]. *CDK5* regulates the development of sperm in the testis, playing an important role in spermatid tail development, spermatid differentiation and spermiogenesis [46]. Our study demonstrated that silencing *PxCDK5* leads to a notable reduction in male fertility among *Plutella xylostella*. Studies have found that *CDK5* exists in spermatocytes in meiotic metaphase, and *PxCDK5* plays an important role in the microtubules in the testis and meiosis of *Plutella xylostella* [47]. Our previous research has also demonstrated that the exposure of male *Plutella xylostella* to irradiation results in distortion of the sperm bundles and sperm flagella [48]. Based on these findings, we hypothesized that irradiation exposure reduces the expression of *PxCDK5* in male *Plutella xylostella*, resulting in the distortion of sperm bundles and flagella, consequently impairing fertility.

In pursuit of enhancing the efficacy of irradiation in pest control, researchers have explored various methodologies integrating irradiation with complementary technologies. Studies have suggested the synergistic utilization of cytoplasmic incompatibility, irradiation and transgenic technologies to control male mosquitoes [49] and *Aedes albopictus* [50,51], thereby capitalizing on their respective strengths to achieve superior control outcomes. The present study integrated irradiation with RNA interference, revealing a pronounced reduction in male *Plutella xylostella* fertility compared to treatments involving irradiation alone or RNA interference alone. Furthermore, the observed delay in fertility recovery within the *Plutella xylostella* population underscored the greater efficacy of the combined approach, indicating a more sustained and potent control mechanism over the population dynamics of *Plutella xylostella*.

## 4. Materials and Methods

### 4.1. Insect Rearing

The pupae of *Plutella xylostella* (Linnaeus) were collected from a cabbage mustard field in South China Agricultural University in Guangdong Province, China. The laboratory population was reared on Chinese flowering cabbage (*Brassica campestris* L. ssp. chinensis var. utilis Tsen et Lee) in a wooden cage (with a length, width and height of 50 cm × 45 cm × 45 cm). The cages were placed in an artificial climate chamber (Zhujiang, Guangdong, China) at 25 ± 1 °C, 60 ± 5% RH and 16L:8D.

### 4.2. ^60^Co-γ Irradiation of Male Pupae

Mature male pupae were exposed to ^60^Co-γ irradiation at a dose rate of 16.67 Gy/min. The ^60^Co-γ irradiation source was provided by Nordion Company in Ottawa, ON, Canada. The irradiation experiment was conducted at Furui High Energy Technology Co., Ltd., located in Guangzhou City, Guangdong Province, China. A total of 600 six-day-old male pupae with the same growth pattern were selected for the experiment. These pupae were individually placed in petri dishes (with a diameter of 100 mm and a height of 15 mm) for irradiation treatment, with three replicates for each treatment.

To investigate the effects on the expression levels of nine genes of irradiation, 100 normal six-day-old male pupae with the same growth pattern were exposed to ^60^Co-γ radiation at 0 Gy, 200 Gy and 400 Gy.

To investigate the dose–effect relationship between irradiation dose and relative gene expression, the six-day-old male pupae of *Plutella xylostella* were exposed to ^60^Co-γ radiation at 0 Gy, 50 Gy, 100 Gy, 150 Gy, 200 Gy, 250 Gy, 300 Gy, 350 Gy and 400 Gy.

To investigate the effect of low-dose irradiation combined with gene-knockdown on male *Plutella xylostella*, the six-day-old male pupae of *Plutella xylostella* were exposed to ^60^Co-γ radiation at 65 Gy and 111 Gy.

### 4.3. Bioassay of the Effects on Fecundity and Vitality

The irradiated males and wild females, alongside wild males and wild females, were allowed to mate naturally for multiple generations. Subsequently, the number of eggs laid and the hatching rates for each generation were recorded. Additionally, parameters, including the pupation rate, the emergence rate and the lifespan of males and females in each generation, were assessed.

#### 4.3.1. Oviposition and Egg Hatchability

Male and female adults were placed together in a clean cage (with a length, width and height of 50 cm × 45 cm × 45 cm) and were allowed to mate freely. Subsequently, after mating, the female adults were transferred to a new cage. To provide sustenance for the female adults, absorbent cotton wool with 10% honey water solution was placed in a petri dish and was replaced daily. A fresh cabbage leaf was suspended in the cage to stimulate oviposition, with the cabbage leaf being replaced every two days. The number of eggs laid on the cabbage leaf was observed and documented using a stereoscope until the natural death of the female adults. The cabbage leaves containing eggs were then transferred into a petri dish, and the hatchability of the eggs was observed and recorded using a stereoscope (Motic SMZ-171, Xiamen, Fujian, China). Every 30 eggs were taken as a group, and three experimental replicates were conducted for each group.

#### 4.3.2. Pupation Rate, Emergence Rate and Adult Lifespan

The mature larvae of *Plutella xylostella* were transferred into a clean cage (with a length, width and height of 50 cm × 45 cm × 45 cm) and were fed with cabbage seedlings until they pupated naturally. Subsequently, the number of pupae was documented and the pupation rate was calculated. Each treatment comprised three replicates and each replicate contained 30 larvae.

The pupae were transferred into a clean cage (with a length, width and height of 50 cm × 45 cm × 45 cm) and were left to naturally emerge. Subsequently, the eclosion number was documented and the eclosion rate was computed. Each treatment comprised three replicates and each replicate contained 30 pupae.

The lifespan of an adult was defined as the duration from emergence to natural death, measured in days. Each treatment comprised three replicates and each replicate contained 30 adults.

### 4.4. Dissection and Observation of Testes

Male adults were selected for dissection 24 h after emergence. They were briefly submerged in alcohol (100%) for 3–5 s, and the specimens were transferred into PBS buffer (pH = 7.4) (Thermo Fisher Scientific, Waltham, MA, USA) to eliminate the residual alcohol. They were promptly dissected in 10 µL of PBS buffer on a glass slide under a stereoscope (Motic SMZ-171, Fujian, China). Subsequently, the buffer was removed through centrifugation at 10,000 rpm and 4 °C for 10 min, and the testes were immediately frozen in liquid nitrogen and stored in a −80 °C refrigerator for subsequent analysis.

### 4.5. Measurements of the Number of Sperm Bundles and the Weight of the Testes

The testes were transferred to a glass slide and were washed with PBS (pH = 7.2). Then, they were transferred to 4% paraformaldehyde fixed liquid and the sperm bundles were washed out and stained. A total of 2 μL of sample solution containing sperm bundles was added to the glass slide and was transferred to a laser confocal microscope (Nikon, Tokyo, Japan) for photographing and observation. The number of two types of sperm bundles was recorded under the laser confocal microscope. The testes were placed into centrifuge tubes (1.5 mL) containing PBS buffer and were weighed using a 10,000th scale (Sartorius, Göttingen, Germany).

### 4.6. Total RNA Extraction and cDNA Synthesis

The total RNA was extracted from individuals or tissues using a Total RNA Extraction Kit (Omega BIO-TEK, Norcross, GA, USA), according to the manufacturer’s instructions. The concentration of RNA was determined using a Nanodrop spectrophotometer (Shimadzu, Kyoto, Japan), and RNA purity was detected using agarose gel electrophoresis. The cDNA template was synthesized using the FastQuant cDNA Synthesis Kit (TIANGEN, Beijing, China).

### 4.7. Quantitative Real-Time PCR (qRT-PCR) Analysis

Nine genes were chosen for qPCR analysis based on their differential expression in the transcriptome libraries. The total RNA was extracted from irradiated males using a Total RNA Extraction Kit (Omega BIO–TEK, Norcross, GA, USA). The primer sequences are shown in Appendix A. *GADPH* was used as the internal reference gene. The reaction was completed in a 20 μL volume. The amplification conditions were as follows: denaturation at 95 °C for 1 min, followed by 39 cycles of 95 °C for 20 s, 60 °C for 20 s and 72 °C for 30 s. The data are expressed as the mean of three independent experiments ± SE (standard error).

### 4.8. RNA Interference, Including dsRNA Synthesis and dsRNA Injecting

Specific primers containing the T7 RNA polymerase promoter sequence were designed for dsRNA synthesis (Appendix A). The PCR program was performed as follows: 98 °C for 3 min; 33 cycles of 98 °C for 10 s, 56–61 °C for 15 s and 72 °C for 30 s; and an additional extension at 72 °C for 5 min. The PCR products were further purified using a Gel Extraction Kit (Omega BIO–TEK, Norcross, GA, USA). The double-stranded RNA (dsRNA) of *PxPI3K*, *PxJAK2*, *PxmTOR*, *PxPTEN*, *PxAKT*, *PxPDK*, *PxCDK5*, *PxNFkB* and *PxIKKα* were synthesized using the T7 RiboMAXTM Express RNAi System (Promega, Madison, WI, USA), following the manufacturer’s recommendations. The dsRNAs of *PxPI3K* (ds*PxPI3K*), *PxJAK2* (ds*PxJAK2*), *PxmTOR* (ds*PxmTOR*), *PxPTEN* (ds*PxPTEN*), *PxAKT* (ds*PxAKT*), *PxPDK1* (ds*PxPDK1*), *PxCDK5* (ds*PxCDK5*), *PxNFkB* (ds*PxNFkB*) and *PxIKKα* (ds*PxIKKα*) were injected into the eighth abdominal segment of six-day-old male pupae using a microinjector (Eppendorf, Hamburg, Germany). Furthermore, six-day-old male pupae injected with diethyl pyrocarbonate (DEPC)-treated water (DEPC water) were used as a negative control (DEPC injection), whereas no treatment was performed for the blank control (CK). The injected individuals were collected at 12 and 24 h after injection for gene silencing efficiency detection.

### 4.9. Statistical Analysis

The 2^−ΔΔ^CT method was used to calculate the relative expression of genes for the qPCR test results. Statistical data from the experiments, including the egg laying volume, hatching rate, pupation rate and emergence rate of *Plutella xylostella*, were subjected to an analysis of variance (ANOVA) using SPSS software version 18.0 (SPSS, Inc., Chicago, IL, USA). Duncan Multiple Range Tests (DMRTs) and Tukey’s range tests were used for significance analysis, and the means were separated at the 0.01%, 0.1% and 5% significance level. The dose–effect relationship between irradiation dose and relative gene expression was fitted using Excel (Microsoft Office version 2016).

## 5. Conclusions

Nine *Plutella xylostella* genes were verified using RNA interference and qRT-PCR analysis. *PxAKT* and *PxCDK5* are key genes that regulate the reproduction of male *Plutella xylostella*. RNA interference was used to knock down *PxAKT* and *PxCDK5* expression in male *Plutella xylostella*, leading to decreases in the ratio of eupyrene sperm bundles to apyrene sperm bundles, the number of eggs produced and the hatching rate. RNA interference technology was used to knock down these two genes, which was combined with low-dose irradiation to investigate the effect on male *Plutella xylostella* fertility. The results showed that the combination of the two treatments had a significantly greater impact on the fertility of *Plutella xylostella* than irradiation treatment or RNAi treatment alone. This study improves our understanding of the genes related to reproduction in *Plutella xylostella* and combines irradiation and RNA interference for developing a new strategy to control *Plutella xylostella*.

The present study suggests that combining irradiation with RNAi technology is more effective in reducing male *Plutella xylostella* fertility compared to either irradiation treatment alone or dsRNA injection alone. Moreover, the fertility of the *Plutella xylostella* population takes longer to recover following combined treatment with irradiation and RNAi technology, indicating that this approach exerts greater harm on the reproductive system.

## Figures and Tables

**Figure 1 ijms-25-04623-f001:**
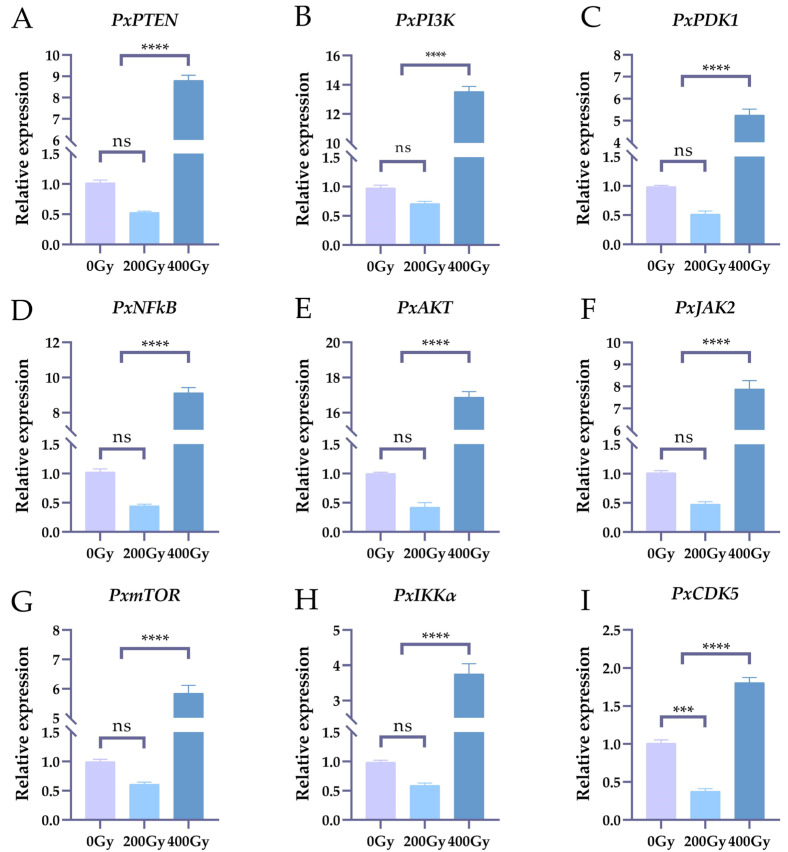
The effects on the expression levels of nine genes after irradiation treatment with 0 Gy, 200 Gy and 400 Gy. (**A**–**I**): The relative expression level of *PxPTEN*, *PxPI3K*, *PxPDK1*, *PxNFkB*, *PxAKT*, *PxJAK2*, *PxmTOR*, *PxIKKα*, *PxCDK5* after irradiation treatment with 0 Gy, 200 Gy and 400 Gy. The error bars display the standard error of the mean based on three biological replicates. According to Tukey’s range test (alpha = 0.05), “ns” denotes no significant difference, “***” denotes significant differences (*p* < 0.001) and “****” denotes significant differences (*p* < 0.0001).

**Figure 2 ijms-25-04623-f002:**
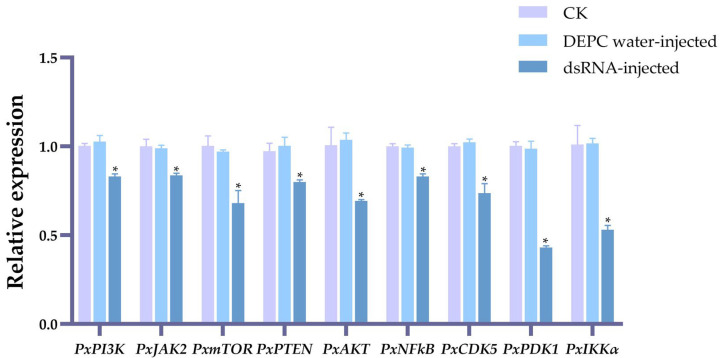
The silencing efficiency of nine genes following dsRNA injection, with CK (no treatment for pests, serving as the blank control group) and DEPC water-injected groups as the control groups. The error bars indicate the standard errors of the means. “*” on the error bar indicates a significant difference among the three treatments (*p* < 0.05), using Duncan’s new multiple range test.

**Figure 3 ijms-25-04623-f003:**
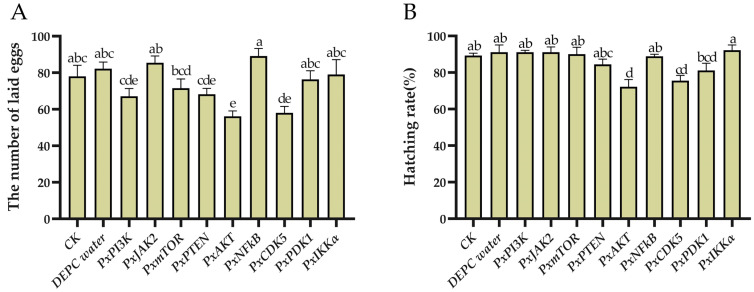
The effects of gene knockdown on *Plutella xylostella* reproduction. (**A**) The number of laid eggs. (**B**) Hatching rate. The error bars indicate the standard errors of the means. Different letters on the error bar indicate significant differences among the treatments, including CK (no treatment for pests, used as the blank control group), DEPC water-injected treatment and dsRNA injection for the nine genes (*p* < 0.05), using Duncan’s new multiple range test.

**Figure 4 ijms-25-04623-f004:**
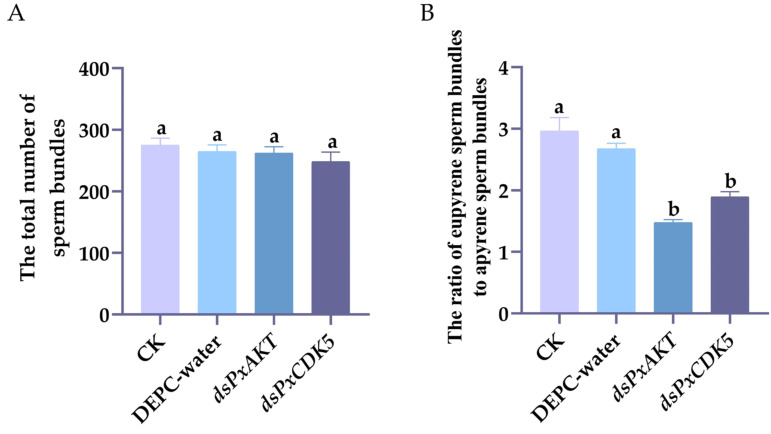
(**A**) The total number of sperm bundles. (**B**) The ratio of eupyrene sperm bundles to apyrene sperm bundles. Different letters on the error bars indicate significant differences between the treatments, including CK (no treatment for pests, used as the blank control group), DEPC water-injected treatment and ds*PxAKT*- and ds*PxCDK5*-injected treatment (*p* < 0.05), using Duncan’s new multiple range test.

**Figure 5 ijms-25-04623-f005:**
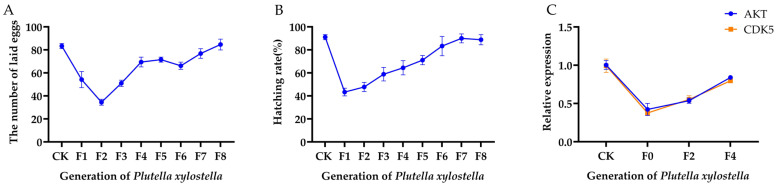
(**A**) The number of eggs laid by *Plutella xylostella* in the F1–F8 generations after 200 Gy irradiation. (**B**) The hatching rate of *Plutella xylostella* in the F1–F8 generations after 200 Gy irradiation. (**C**) The expression levels of *PxAKT* and *PxCDK5* in the F0–F4 generations after 200 Gy irradiation.

**Figure 6 ijms-25-04623-f006:**
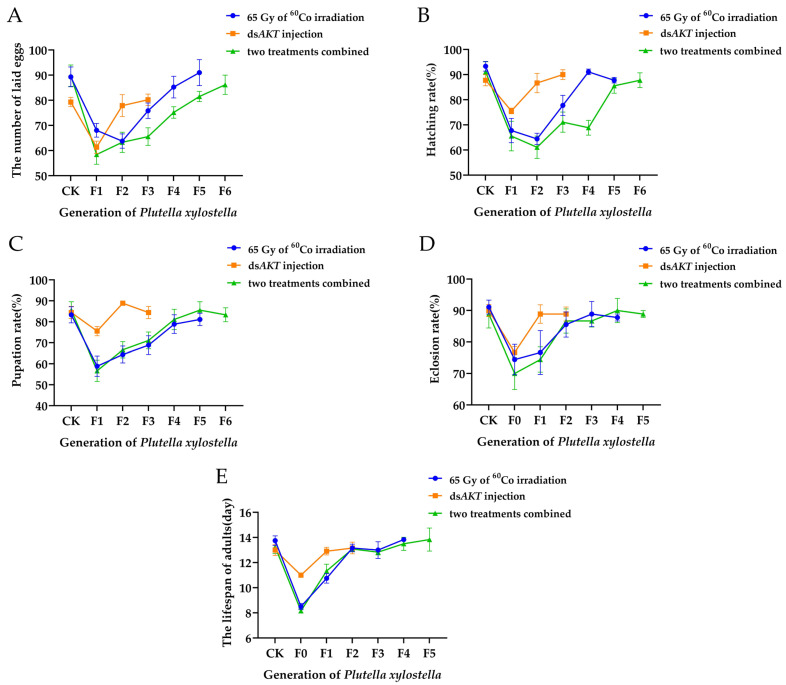
The impact of the three treatments, including 65 Gy of ^60^Co irradiation, ds*PxAKT* injection and both combined, on *Plutella xylostella* fertility. (**A**) The number of eggs laid by F1 to F6 generations after three treatments. (**B**) The hatching rate of F1 to F6 generations after three treatments. (**C**) The pupation rate of F1 to F6 generations after three treatments. (**D**) The eclosion rate of F1 to F5 generations after three treatments. (**E**) The lifespan of adults from F1 to F5 generations after three treatments.

**Figure 7 ijms-25-04623-f007:**
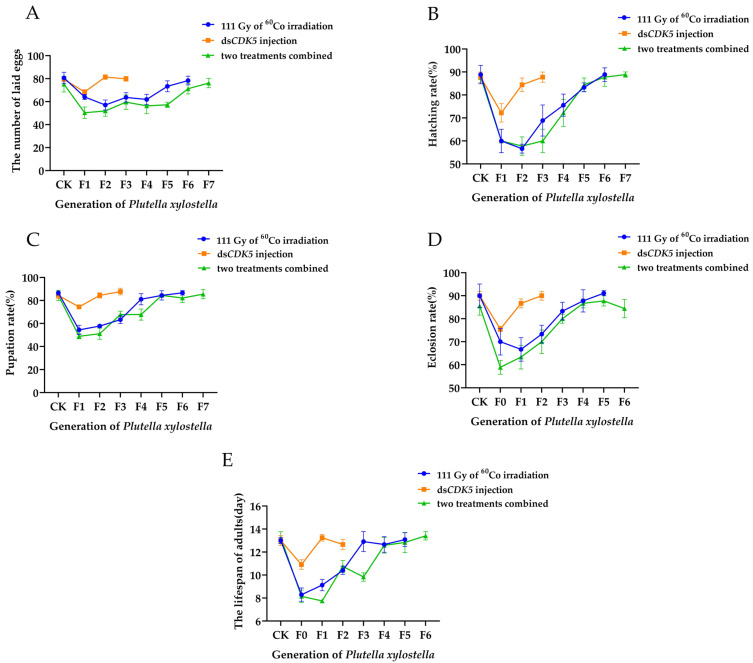
The impact of three treatments, including 111 Gy of ^60^Co irradiation, ds*PxCDK5* injection and both combined, on *Plutella xylostella* fertility. (**A**) The number of eggs laid by the F1 to F7 generations after three treatments. (**B**) The hatching rate of F1 to F7 generations after three treatments. (**C**) The pupation rate of F1 to F7 generations after three treatments. (**D**) The eclosion rate of F1 to F6 generations after three treatments. (**E**) The lifespan of adults from F1 to F6 generations after three treatments.

## Data Availability

The original contributions presented in the study are included in the article/Appendix A, further inquiries can be directed to the corresponding author.

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
