# Peer review of "Verification of AKT and CDK5 Gene and RNA Interference Combined with Irradiation to Mediate Fertility Changes in Plutella xylostella (Linnaeus)"

_ijms, 2024, doi:10.3390/ijms25094623_

Round 1

Reviewer 1 Report

Comments and Suggestions for Authors

In present work authors examined the effect of  RNA interference combined and irradiation and both techniques on fertility changes on Plutella xylostella (Linnaeus) with particular interest in AKT and CDK5 genes.

Manuscript is well-written, experiments well-described. The weakness of present work is poor Methodology description; there is lack, for example:

-information how particular treatments impact the emergence or lifespan of insect;

-information about controls used in experiments;

- information about the methodology of injection with dsRNA;

- information about statistics;

Please add this information to the Methodology section. 

Reviewer 2 Report

Comments and Suggestions for Authors

The study carried out trying to determine the effect of irradiation effects and RNA interference at the reproductive level and characterization of selected related genes in Plutella Xylostella is of interest. The research seems to be well performed and with apparent good results. However, the manuscript requires major changes:

- The methodology should be completed. For example, the irradiation levels are not described, the RNAi treatment is very lightly described, and there are groups such as the DEPC water-injected group that are not detailed either. There is also no section that details the applied statistics. This section requires a significant improvement to be able to reproduce the study.

- At the background level of bibliographic references it should also be improved. There are many statements without including bibliographic references. And at the level of Introduction and Discussion it does not seem sufficiently developed for publication.

- The results must be refined, indicating differences at a quantitative level. Its structure must correspond to that followed by the methodology. Significant differences that are not shown in all of them should be indicated in the graphs. Some legends need also to be improved.

I recommend that the manuscript be reviewed by a native English professional to refine the sentences to submit a manuscript that is as refined as possible.

A document with more detailed comments is attached.

Comments on the Quality of English Language

English require major changes.

Author Response

Please write down "Please see the attachment.

Round 2

Reviewer 2 Report

Comments and Suggestions for Authors

In my opinion, the manuscript has been significantly improved for publication, including most of my previous suggestions. I only noticed some minor changes:

- Please quantify the differences in the abstract.

- Line 51: Plutella xylostella' seems redundant. 

- Letters in figures should be standardised to 'Palatino Linotype'.

- Complete statistics in Figure 5 and Figure 6. Some figures should be enlarged for easier reading.

- I think the discussion should be improved to include more background.

- Line 473: Please include DEPC abbreviation.

- Line 481: Please, include city and country por SPSS software.

- Details of statistical ANOVA should be provided and methodological details should be completed in 4.9 Statistical analysis. In the results, a more precise significance analysis than the 95% indicated in 4.9 is sometimes given.

Comments on the Quality of English Language

I think only minor changes are required.
